# Structural violence and institutionalized individuals: A paleopathological perspective on a continuing issue

**Carlina de la Cova**[1], **Madeleine Mant**[2]*, **Megan B. Brickley**[3]

**1** Department of Anthropology, University of South Carolina, Columbia, South Carolina, United States of America, **2** Department of Anthropology, University of Toronto Mississauga, Mississauga, Ontario, Canada, **3** Department of Anthropology, McMaster University, Hamilton, Ontario, Canada

* maddy.mant@utoronto.ca

## Abstract

Past and present institutions (e.g., state and public hospitals, assisted living facilities, public nursing homes) have struggled with structural issues tied to patient care and neglect, which often manifests in the form of fracture trauma, and may explain why institutionalized individuals are at higher risk for this injury. Six hundred individuals from the Robert J. Terry Anatomical Collection born between 1822–1877 were examined to investigate hip fracture prevalence. Analysis of associated records and documentary data, including death, morgue, and census records, revealed that 36.3% (n = 218) of these individuals died in institutions such as the St. Louis State Hospital, City Infirmary, and Missouri State Hospital No. 4. Of the institutionalized individuals, 4.3% had evidence of hip fracture, significantly higher than the non-institutionalized (2.3%). Records revealed that many hip fractures were suffered around the time of death in state hospitals and were preventable, resulting from structural issues tied to understaffing and underfunding. Forensic and clinical literature, as well as current news media, indicate that structural violence in the forms of underfunding and understaffing continues to manifest as hip fractures harming institutionalized individuals today. This paper demonstrates how an anthropological perspective using paleopathological analysis sheds light on the chronicity and time depth of this issue, with the aim of driving public policy to entrench the equitable care of institutionalized people as a human right.

## Introduction

Institutional settings, defined in this paper as state hospitals, assisted living facilities, state schools, public hospitals, and nursing homes, have a history of struggling with patient care and neglect, which persists into the present day. By the late 20th century, the push to deinstitutionalize resulted in the closing of many decrepit state hospital and asylum complexes in the United States and Western Europe [1–3]. Ultimately, nursing homes, smaller clinics, and assisted living facilities arose to house and care for the elderly and mentally ill [4–6]. In recent years these facilities have undergone critical evaluations, revealing that many are plagued with the same issues as their state-based predecessors; current news about modern day healthcare facilities and assisted living facilities provide evidence of ongoing structural issues [e.g., 7–9].

**Data Availability Statement:** There are partial restrictions on the data as research protocols associated with the Robert J. Terry Anatomical Skeletal collections have changed since the data for

this study was gathered. Since the start of the COVID-19 pandemic, the Smithsonian has stopped further research with the Terry Collection and the Hamann-Todd Collection has shifted to require research approval by the collections manager that demonstrates ethical engagement with the collections and the peoples that comprise them. Questions regarding further research with the collections should be directed to the Biological Anthropology team at the Smithsonian, using the Human Remains Research Approval form (https://naturalhistory.si.edu/research/anthropology/collections-and-archives-access). Summarized and anonymized data may be shared with individual researchers upon reasonable request.

**Funding:** This research was funded by Indiana University (CDC), a New Faculty research grant from the University of North Carolina at Greensboro (CDC), the Provost's Office of the University of South Carolina (CDC), the Smithsonian Institution (CDC), and the Canada Research Chairs program (#231563, MBB). The funders had no role in study design, data collection and analysis, decision to publish, or preparation of the manuscript.

**Competing interests:** The authors have declared that no competing interests exist.

Structural issues, defined in this paper as reduced funding, poor maintenance, and under-staffing of institutions, result in neglect and harm of patients, which can manifest in the form of structural and physical violence, with resultant fractures derived from direct trauma and underlying pathological conditions [10, 11]. Structural violence is a form of violence in which social structures "suppress agency and prevent individuals, groups, and societies from reaching their social, economic, and biological potential" [12:31]. Despite governmental codes regulating care in modern institutions, individuals confined to these places, especially the elderly, remain at higher risk for hip fracture when compared to the non-institutionalized [13, 14]. Furthermore, some scholars have indicated that admission to an institutional setting, such as a nursing home, increases hip fracture risk, with trauma occurring in the early months of admission and dependent on the level of available patient care [14, 15]. Modern institutionalized individuals with hip fractures also have worse functional outcomes following fracture than their contemporaries in community-level settings [e.g., 16–18]. These circumstances reflect settings of structural violence, as the factors are "embedded in the political and economic organization" and "are violent because they cause injury to people" [19:1686].

Bioarchaeology is ideally positioned to assist clinicians, researchers, and public policy makers in comprehending how these problems both have a long history and persist today. Human skeletal remains hold a rich information store when approached with care and considering the limitations of assessing behaviors and health [20–22]. In recent years, bioarchaeological research, and more specifically paleopathology (the study of diseases and related conditions in the past using human remains), paired with historical contextual sources, has shed light on the prevalence of hip fractures amongst the historically institutionalized through skeletal and archival analyses, illustrating the long-standing relationship between structural violence, poor institutional conditions, patient neglect, overcrowding, and fractures [10, 23–29]. This paper will build on these works by demonstrating how an anthropological analysis, through the lens of paleopathology, can reveal the time depth of the troubling ties between structural violence, hip trauma, and institutionalization in both past and present institutionalized settings. By uncovering how structural challenges have resulted in consistent patterns of harm to the individuals that institutions are meant to protect, is it hoped that this research can help inform future changes to protect vulnerable institutionalized individuals.

## Individuals examined and methods

### The Robert J. Terry Anatomical Skeletal Collection

The Robert J. Terry Anatomical Skeletal Collection (RJTC) is a documented series that contains the remains of 1,728 adult individuals (>18 years) who died in St. Louis and the broader state of Missouri between 1910 and 1967 [30]. The collection was initially amassed by Robert J. Terry, professor and anatomy chair at Washington University, St. Louis in 1910. Contemporary Missouri anatomical legislation (which still applies today) indicated that the unclaimed poor, including those in state institutions, were to be non-consensually offered to medical and mortuary schools for anatomization [11]. Terry aided in the modification of these acts, narrowing the interval for claiming the dead and increasing the period educational institutions held bodies (30 days) before they were anatomized [31]. Relations of persons that died in state-funded institutions, including poor farms, city hospitals, infirmaries, state mental hospitals, and prisons, or were to be buried at public expense, had 36 to 72 hours to claim their loved ones. It was these legalized means that provided Terry with the vehicle to collect and curate the skeletons of the unclaimed poor for personal research and teaching purposes until his retirement in 1941. Terry's colleague Mildred "Trot" Trotter, one of the nation's first female physical

anthropologists, continued amassing his collection until her retirement in 1967; it was then placed on loan to the Smithsonian Museum of Natural History, where it remains today [32].

The RJTC is a documented series, which means that age, biological sex, socially ascribed race, stature, date of death, cause of death, place of death, biological measurements, and other biographical or pathological data was recorded for most individuals in the collection at their time of death, or upon entry into the collection, either by morgue records or additional research conducted by Terry [30, 33]. Terry's personal letters reveal that he was meticulous in obtaining biographical and pathological data for each individual, often communicating with associated hospitals of death to ensure each person was properly documented for effective research use [33]. Historical research on the RJTC by de la Cova [23], using death certificates, newspapers, and Terry's personal correspondence, has indicated that most individuals died in public and charity hospitals, such as City Hospital #1 and #2, the City Infirmary (or poorhouse), and mental institutions from across Missouri, including the St. Louis State Hospital and State Hospitals Nos. 1 through 4. Whilst legislation provided Terry with easy and legal accessibility of human bodies for research, these individuals did not consent to anatomization and curation; it was mandated by the state due their social status. Thus, structural violence played, and continues to play, a role in regard to how the poor, homeless, marginalized, and mentally ill are treated in both life and death.

## Methods

A sample of 600 males and females born between 1822 and 1877 (Table 1) from the RJTC were examined to determine the relationship between structural violence, apathy, institutionalization, and trauma, via the evaluation of hip fractures. All individuals born from 1822 to 1877 were selected for analysis as they lived through the periods in which state hospitals and mental institutions were founded, experienced their peak patient intakes in the early 20th-century, and ultimately declined in the mid-20th century.

Skeletal remains of the individuals studied were examined macroscopically. Sex was estimated by standard osteological and sexually dimorphic features of the os coxa and skull; age at death was estimated from observations of the pubic symphysis and auricular surfaces [34, 35]. Skeletal results were then compared with the documentary evidence available for each individual (e.g., morgue and death records). Previous studies have found discrepancies between the results of osteological analyses and evidence such as autopsy reports [36], emphasizing the importance of drawing together multiple lines of data. For this study, multiple lines of data based on historical and osteological research, were utilized to ensure that a complete understanding of the origins and composition of the sample as well as individual hip fractures, were accurately assessed.

Table 1. Sample distribution by social race and biological sex.

|  | n | percent |
|---|---|---|
| African American | 217 | 36.2 |
| Euro-American | 383 | 63.8 |
| Female | 243 | 40.5 |
| Male | 357 | 59.5 |
| African American female | 104 | 17.3 |
| African American male | 113 | 18.8 |
| Euro-American female | 139 | 23.2 |
| Euro-American male | 244 | 40.7 |

To qualify as institutionalized, historical research of associated death, mortuary, and census records were examined to ascertain the institutionalized status of each individual in the sample. These death records had to clearly indicate that individuals died in the city infirmary, a state mental hospital, a nursing home, and/or been hospitalized for more than 45 days. Institutionalization was recorded as Yes or No based on these criteria. The specific institutions of death were also recorded to determine if fractures were concentrated in certain institutions, which could imply underlying factors tied to structural violence.

All available and observable left and right femora were examined; all individuals had both femora present. Hip fracture trauma was evaluated on all available and observable femora using methods described by Lovell [37], Mant et al. [25], and de la Cova [10, 11]. Fractures were recorded as present or absent. If perimortem (occurring around the time of death) hip trauma was suspected, fracture margins were examined using forensic anthropological criteria for perimortem fractures, including assessing the presence or absence of: smooth fracture surfaces, continuous bone colour from cortical surface to fractured ledge, lack of healing, radiating hairline fractures, and hinging of bone fragments [38–40]. Based on these criteria each individual was noted as either "present" or "absent" for the presence of a perimortem hip fracture.

Hip fracture trauma was chosen given its clinical significance and prevalence amongst present-day institutionalized individuals. As the RJTC is a documented collection, associated morgue records were further examined for individuals with unhealed hip fractures to determine if this trauma occurred perimortem and was a contributing factor in death. This was done through the examination of morgue records, anthropometric data sheets Terry kept for each individual, and death certificates. If all records were in agreement that an individual had a perimortem fracture in the location observed, and death records indicated that the fracture contributed to the individual's cause of death, the fracture was classified as perimortem.

Author CDC undertook the initial osteological observations and documentary research; authors MM and MBB analyzed and crosschecked the fracture results with the death certificates to ensure all individuals identified as having a perimortem fracture aligned with the available documentary evidence. While research ethics board approval was not required for this research, the authors focused upon uniting individuals assessed with their documentary evidence. Given the use of the RJTC historically [23], this research and manuscript seeks to not only humanize the individuals in the collection but restore their voices and shed light on their lived and postmortem experiences.

Individuals were placed in institutionalized and non-institutionalized cohorts. Frequency, chi-squared (p<0.05), and odds ratio analyses were conducted using SPSS V22.0 to determine if significant differences in hip fracture trauma existed between institutionalized and non-institutionalized individuals.

## Results

The average age of the sample was 70.1 years of age, with most individuals falling between the ages of 60 to 80 (Fig 1). The sample was further broken into ten-year cohorts to better assess age distribution and revealed that most individuals, or a total of 84.1% of the sample, fell between 60 to 89 years of age, with 32.8% of the sample being between 60 to 69 years of age, 35.3% of the sample being between 70 to 79 years of age and 16.0% of the sample 80 to 89 years of age (Table 2).

Further age analyses broke the sample down into those institutionalized versus those not institutionalized to determine what the average ages and age ranges were in each group. The average age of those not institutionalized was 68.0 (Fig 2); the average age of those

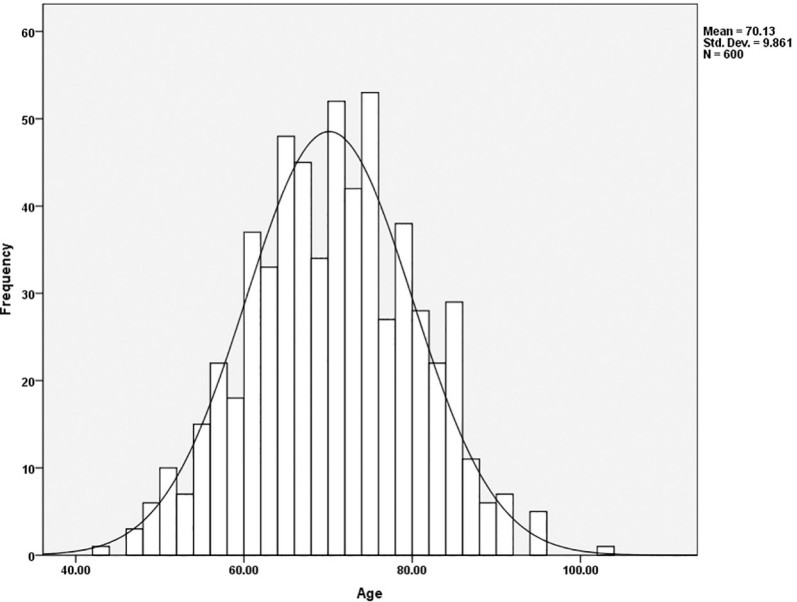

**Fig 1. Age distribution of the entire sample.**

institutionalized was 73.9 (Fig 3). However, age distributions revealed that the majority of both samples were comprised of individuals between 50 to 89 years (Table 3). For those not institutionalized, 15.7% of the sample (n = 60) were 50 to 59 years of age. This was followed by 37.2% (n = 142) associated with the age ranges of 60–69, 32.5% (n = 124) between 70 and 79, and 11.3% (n = 43) comprising the years 80 to 89. The distribution of age cohorts amongst the institutionalized included only 12 individuals (5.5%) in the 50 to 59 age ranges, 25.2% (n = 55) aged 60 to 69, 40.4% (n = 88) between the ages of 70 to 79, and 24.3% (n = 53) in the 80 to 89 age range. This suggests that institutionalized individuals are slightly older, as indicated by the mean age differences, but the majority of the sample is still found in the same age cohorts for both non-institutionalized and institutionalized individuals.

218 individuals, or 36.2% of the sample, had been institutionalized (Table 4). Of those 25.3% were Euro-American, and more specifically, 15.3% were white women. Thus, white females had the highest rates of institutionalization in the sample; African American males had the lowest.

Analyses of the cohorts and hip trauma revealed that 6.7% of the total sample suffered a hip fracture (Table 5). There were no bilateral fractures. Females had significantly higher rates of

**Table 2. Age distribution of sample.**

| Cohort age | n | percent |
|------------|------|---------|
| 40–49 | 10 | 1.7 |
| 50–59 | 72 | 12.0 |
| 60–69 | 197 | 32.8 |
| 70–79 | 212 | 35.3 |
| 80–89 | 96 | 16.0 |
| 90+ | 13 | 2.2 |

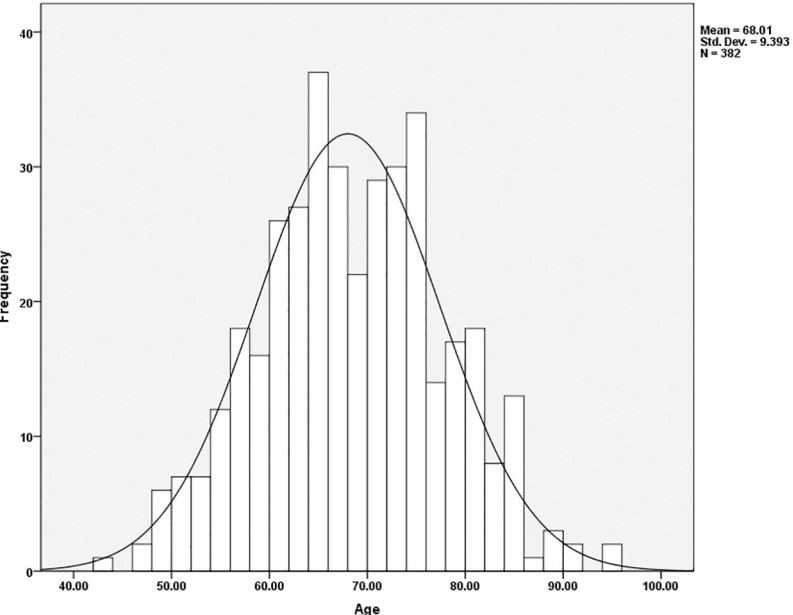

**Fig 2. Age distribution of non-institutionalized individuals in the sample.**

this trauma (p = 0.000), with 5.2% of the sample affected. Amongst the females, white women (4.3%) had significantly more instances of hip fractures (Table 5). In regard to age cohorts, due to low counts, the 40–49 and 50–59 cohorts were combined into a new cohort, 40–59. The 80–89 and 90–99 age cohorts were also combined into an 80+ cohort. When these groups were analyzed, significant differences existed (p = .008), with the 40–59 age cohort having the lowest rates of hip trauma (n = 2, 0.3%) as indicated in Table 5. It should be noted that in the 60–69

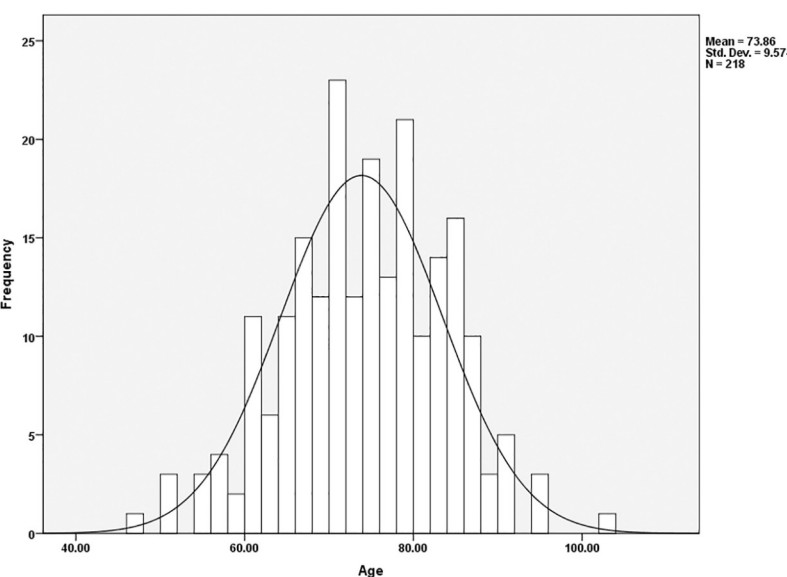

**Fig 3. Age distribution of institutionalized individuals in the sample.**

**Table 3. Age distribution of the sample in regard to institutionalized status.**

| | Not institutionalized | | Institutionalized | |
|---|---|---|---|---|
| Cohort age | n | Percent | n | percent |
| 40–49 | 9 | 2.4 | 1 | 0.5 |
| 50–59 | 60 | 15.7 | 12 | 5.5 |
| 60–69 | 142 | 37.2 | 55 | 25.2 |
| 70–79 | 124 | 32.5 | 88 | 40.4 |
| 80–89 | 43 | 11.3 | 53 | 24.3 |
| 90+ | 4 | 1.0 | 9 | 4.1 |
| Total | 382 | 100.0 | 218 | 100.0 |

cohort, 11 individuals (1.8%) sustained a hip fracture. Two percent of the sample in the 70–79 age cohort and 15 individuals or 2.5% in the 80+ age category had hip fractures.

An examination of the institutionalized cohorts revealed significantly higher frequencies of hip fractures in the institutionalized sample (4.3%, p = .000) when compared to those not institutionalized (2.3%) (Table 6). Further analyses were undertaken to ascertain the demographics of this trauma amongst the institutionalized. Only institutionalized individuals were examined in these tests. Results indicated that significantly more women (11.5%, p = .000) suffered a hip fracture when compared to men (0.5%), with only one male affected.

Whilst Euro-Americans were more afflicted with hip trauma than African Americans (10.1% of the sample versus 1.8%), these results were not statistically significant. Combined social race and biological sex cohorts indicated that, overwhelmingly more white institutionalized females suffered hip fractures when compared to the other cohorts. Of the 26 institutionalized individuals that suffered hip trauma, 21 were white females. Although the results of this test were significant, as was the likelihood ratio, the interpretive power is limited due to the low cell counts associated with the chi-squared analysis. However, that the majority of the hip fractures observed in the institutionalized occurred amongst white females cannot be ignored. Four persons (1.8%) aged 60 to 69, eight (3.7%) in the 70–79 cohort, and fourteen (6.4%) aged 80+ suffered hip fractures.

Risk estimates were also performed to determine the relationship between hip fractures and the institutionalized. Based on the data of this study individuals not institutionalized had a decreased risk for hip fracture (Table 7). The sample was then divided into two age cohorts

**Table 4. Terry sample institutionalized distribution.**

| | No (%) | Yes (%) | Total (%) |
|---|---|---|---|
| Institutionalized | 382 (63.7) | 218 (36.3) | 600 (100) |
| | No (%) | Yes (%) | Total (%) |
| African American | 151 (25.2) | 66 (11.0) | 217 (36.2) |
| Euro-American | 231 (38.5) | 152 (25.3) | 383 (63.8) |
| Female | 110 (18.3) | 133 (22.2) | 243 (40.5) |
| Male | 272 (45.3) | 85 (14.2) | 357 (59.5) |
| African American female | 63 (10.5) | 41 (6.8) | 104 (17.3) |
| African American male | 88 (14.7) | 25 (4.2) | 113 (18.8) |
| Euro-American female | 47 (7.8) | 92 (15.3) | 139 (23.2) |
| Euro-American male | 184 (30.7) | 60 (10.0) | 244 (40.7) |

**Table 5. Hip fracture analyses of the entire Terry sample.**

| | Absent | Present | Total | | |
|---|---|---|---|---|---|
| Hip fracture frequencies in entire sample | | | | | |
| | Absent | Present | Total | | |
| | 560 (93.3) | 40 (6.7) | 600 | | |
| Chi-Square analyses of hip fractures | | | | | |
| | Absent (%) | Present (%) | Total | $x^2$ | Sig. |
| Female | 212 (35.3) | 31 (5.2) | 243 (40.5) | 24.348 | 0.000 |
| Male | 348 (58.0) | 9 (1.5) | 357 (59.5) | | |
| African American | 208 (34.7) | 9 (1.5) | 217 (36.2) | 3.467 | 0.063 |
| Euro-American | 352 (58.7) | 31 (5.2) | 383 (63.8) | | |
| African American female | 99 (16.5) | 5 (0.8) | 104 (17.3) | 43.089 | 0.000 |
| African American male | 109 (18.2) | 4 (0.7) | 113 (18.8) | | |
| Euro-American female | 113 (18.8) | 26 (4.3) | 139 (23.2) | | |
| Euro-American male | 239 (39.8) | 5 (0.8) | 244 (40.7) | | |
| 40–59 | 80 (13.3) | 2 (0.3) | 82 (13.7) | 11.890 | 0.008 |
| 60–69 | 186 (31.0) | 11 (1.8) | 197 (32.8) | | |
| 70–79 | 200 (33.3) | 12 (2.0) | 212 (35.3) | | |
| 80+ | 94 (15.7) | 15 (2.5) | 109 (18.2) | | |

**Table 6. Hip fracture analyses of Terry individuals that were institutionalized.**

| | Absent | Present | Total | | | |
|---|---|---|---|---|---|---|
| Hip fracture frequencies in entire sample | | | | | | |
| | Absent | Present | Total | | | |
| | 560 (93.3) | 40 (6.7) | 600 | | | |
| Hip fracture frequencies amongst only those institutionalized | | | | | | |
| | Absent | Present | Total | | | |
| | 192 (88.1) | 26 (11.9) | 218 | | | |
| Chi-Square analyses of hip fractures and institutionalization | | | | | | |
| | Absent (%) | Present (%) | Total (%) | $x^2$ | Sig. | Likelihood ratio (Sig.) |
| Not Institutionalized | 368 (61.3) | 14 (2.3) | 382 (63.7) | 15.225 | 0.000 | 13.926 (.000) |
| Institutionalized | 192 (32.0) | 26 (4.3) | 218 (36.3) | | | |
| Female | 108 (49.5) | 25 (11.5) | 133 (61.0) | 15.328 | 0.000 | 19.918 (.000) |
| Male | 84 (38.5) | 1 (0.5) | 85 (39.0) | | | |
| African American | 62 (28.4) | 4 (1.8) | 66 (30.3) | 3.101 | 0.078 | 3.466 (.063) |
| Euro-American | 130 (59.6) | 22 (10.1) | 152 (69.7) | | | |
| African American female | 37 (17.0) | 4 (1.8) | 41 (18.8) | 19.987[a] | 0.000 | 24.115 (.000) |
| African American male | 25 (11.5) | 0 (0.0) | 25 (11.5) | | | |
| Euro-American female | 71 (32.6) | 21 (9.6) | 92 (42.2) | | | |
| Euro-American male | 59 (27.1) | 1 (0.5) | 60 (27.5) | | | |
| 40–59 | 13 (6.0) | 0 (0.0) | 13 (6.0) | 10.268[b] | 0.016 | 10.819 (.013) |
| 60–69 | 51 (23.4) | 4 (1.8) | 55 (25.2) | | | |
| 70–79 | 80 (36.7) | 8 (3.7) | 88 (40.4) | | | |
| 80+ | 48 (22.0) | 14 (6.4) | 62 (28.4) | | | |

[a] 2 cells (25.0%) have expected count less than 5. The minimum expected count is 2.89.

[b] 1 cells (12.5%) have expected count less than 5. The minimum expected count is 1.50.

**Table 7. Odds ratios for institutionalized hip fracture presence.**

| | | Hip Fracture | | |
|---|---|---|---|---|
| | | Absent | Present | Total |
| Institutionalized | No | 369 (61.5) | 15 (2.5) | 384 (64) |
| | Yes | 191 (31.8) | 25 (4.2) | 216 (36) |
| | Total | 560 (93.3) | 40 (6.7) | 600 (100) |
| | | Risk Estimate | | |
| | | | 95% Confidence Interval | |
| | | Value | Lower | Upper |
| Odds Ratio for Institutionalized (no/yes) | | 3.220 | 1.658 | 6.252 |
| For cohort hip fracture = Absent | | 1.087 | 1.031 | 1.145 |
| For cohort hip fracture = Present | | .338 | .182 | .626 |
| N of Valid Cases | | 600 | | |

due to a low sample distribution between the ages of 40 to 59. Individuals were grouped into a 40–79 age cohort and an 80+ cohort to determine if there was an increased risk of hip fracture amongst these ages in institutionalized versus non-institutionalized individuals (Table 8). For those not institutionalized, there appears to be an increased risk for hip fractures in the 40–79 age cohort versus a lower risk in the 80+ cohort. The reverse pattern was observed in the institutionalized cohort.

**Table 8. Odds ratios for age cohorts and hip fractures examined by institutionalized status.**

| | Not Institutionalized | | |
|---|---|---|---|
| | Hip Fracture | | |
| Age Cohort | Absent | Present | Total |
| 40–79 | 322 (84.3) | 13 (3.4) | 335 (87.7) |
| 80+ | 46 (12) | 1 (0.3) | 47 (12.3) |
| Total | 368 (96.3) | 14 (3.7) | 382 (100) |
| Risk Estimate | | | |
| | | 95% Confidence Interval | |
| | Value | Lower | Upper |
| Odds Ratio for Age Cohorts (40–79 / 80+) | .538 | .069 | 4.213 |
| For cohort hip = Absent | .982 | .937 | 1.030 |
| For cohort hip = Present | 1.824 | .244 | 13.625 |
| N of Valid Cases | 382 | | |
| Institutionalized | | | |
| Age Cohort | Absent | Present | Total |
| 40–79 | 144 (66.1) | 12 (5.5) | 156 (71.6) |
| 80+ | 48 (22.0) | 14 (6.4) | 62 (28.4) |
| Total | 192 (88.1) | 26 (11.9) | 218 (100) |
| Risk Estimate | | | |
| | | 95% Confidence Interval | |
| | Value | Lower | Upper |
| Odds Ratio for Age Cohorts (40–79 / 80+) | 3.500 | 1.515 | 8.086 |
| For cohort hip fracture = Absent | 1.192 | 1.035 | 1.374 |
| For cohort hip fracture = Present | .341 | .167 | .695 |
| N of Valid Cases | 218 | | |

**Table 9. Presence and distribution of 17 observed perimortem fractures.**

| | n | Percent of total perimortem hip fractures |
|---|---|---|
| Euro-American perimortem hip fractures | 16 | 94.1 |
| • 14 females | | |
| • 2 males | | |
| African American perimortem hip fractures | 1 | 5.9 |
| • 1 female | | |
| *Age distribution of perimortem hip fractures* | | |
| 40–59 | 0 | 0 |
| 60–69 | 4 | 23.5 |
| 70–79 | 6 | 35.3 |
| 80+ | 7 | 41.2 |
| Institutionalized | 14 | 82.4 |
| Not institutionalized | 3 | 17.6 |

Further examination of the hip fracture findings revealed that of the 40 instances of trauma, 17 (42.5%) were perimortem and had accompanying documentary data that confirmed this (Table 9). Sixteen were Euro-American, comprised of 14 females and two males; only one was an African American female. Age-wise, four individuals between 60 to 69, six in the 70–79 cohort, and seven aged 80+ were affected with hip fractures. Fourteen of the 17 (82.4%) perimortem fractures occurred among institutionalized individuals.

When morgue and death records of the sample were examined with respect to the institution where fracture and death occurred (Fig 4), the City Infirmary had the highest prevalence of hip fractures, followed by the St. Louis State Hospital, which continues to operate in the present day, City Hospital #1 (which catered to poor indigent whites), Pinecrest (a

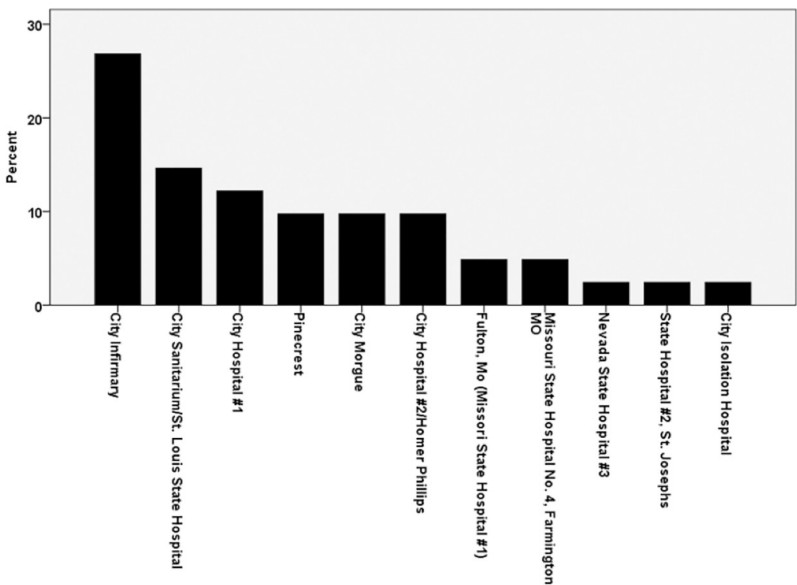

**Fig 4. Institutions associated with hip fractures.**

nursing home facility), the City Morgue, segregated City Hospital #2 for St. Louis's indigent Black population, Fulton State Hospital (which is also still in operation, housing the state's "criminally insane"), and Missouri State Hospitals Nos. 4, 3, and 2. Contemporaneous newspapers, periodicals, and Missouri state government documents were consulted to determine what the living and operating conditions of these institutions were like. Descriptive results from these documents are presented in detail in the discussion section as they reveal the problematic structural issues with which Missouri's state hospitals struggled and how they not only affected the individuals in this study, including overcrowding, lack of doctors and attendants, physical abuse, and poor facility maintenance, but increased fracture risk.

## Discussion

The purpose of this paper was to illustrate, using an anthropological analysis that relied on paleopathological examination, the troubling time-depth between hip trauma, structural violence, and institutionalization in past and present groups. Analysis of individuals born between 1822 and 1877 in the RJTC, a documented skeletal collection, revealed that the institutionalized had higher rates of hip fractures, including perimortem hip fractures, when compared to the non-institutionalized. Recent studies of individuals institutionalized in Cleveland State Hospital and Cleveland's City Infirmary in the early 1900s to 1938, associated with the Hamann-Todd Human Anatomical Collection, share a similar pattern with the presence of perimortem hip fractures and healing radius and rib fractures, which resulted from structural issues within these institutions [10, 11, 28, 29]. Others have demonstrated evidence of violence (structural and inter-personal), with resultant fractures derived from direct trauma and underlying pathological conditions at 19[th]-century institutions across Europe [e.g., 24, 26]. Four individuals dating from 1891 to 1936 buried in the Catholic Cemetery at Meerenberg Psychiatric Hospital in Bloemendaal, the Netherlands, suffered from fractures consistent with osteomalacia, resulting from insufficient exposure to sunlight and a diet low in Vitamin D [24]. Analyses of the remains of individuals associated with correctional facilities/workhouses in 19[th]- and 20[th]-century Switzerland demonstrated high rates of healing fractures, suggesting evidence of interpersonal violence whilst institutionalized, including rib, scapular, and mandibular trauma [26]. These findings indicate the time depth of this consistent and troublesome trend of increased fracture risks amongst those institutionalized in hospitals, nursing homes, and assisted living communities, echoing findings illustrated in clinical studies conducted over the past 30+ years [e.g., 13, 14, 41].

Key and disturbing findings in the current study are not only the higher rates of hip fractures amongst the institutionalized, but the higher rates of perimortem hip trauma amongst this group. Furthermore, another important finding in this research is the statistically significant higher risk of hip fracture for institutionalized individuals in the 80+ age cohort than those not institutionalized. Older age is associated with decreased bone density (osteopenia) and senescence, issues with balance and an overall increased risk of falls [42, 43]; thus the higher number of individuals in the 80+ age cohort overall suffering hip fractures is an expected result. However, of greater concern is the difference in risk for hip fracture between the institutionalized vs. non-institutionalized cohorts; clearly more than just age was a factor in these incidents. This is further emphasized by the higher occurrence of perimortem hip fractures amongst the institutionalized. Certainly, older individuals are at risk for hip fractures [44], but when they are predominantly occurring in higher rates in institutionalized individuals, factors beyond age may be creating unhealthy conditions that are amplifying the risk for hip fracture.

Examination of death records associated with these individuals, particularly those with perimortem fractures revealed how compounding structural issues resulted in fatal fracture consequences amongst the institutionalized persons studied in this work. A discussion of how these issues, when combined with skeletal, contextual, and historical data can be utilized to demonstrate not only the perilous relationship between these factors for the institutionalized, but how they persist in the present day and continue to harm this vulnerable group.

The individuals examined in this study were confined to hospitals during an era when state-funded mental hospitals, infirmaries, and poor houses for the elderly were underfunded, overcrowded, and understaffed. Journalist Albert Maisel shed light on the horrors, abuse, and neglect experienced by patients institutionalized in America's state hospitals in the 1940s. His *LIFE* magazine expose, "Bedlam, 1946", drew upon staff interviews and court documents to reveal the horrid conditions of the institutionalized. He argued that "public neglect and legislative penny-pinching" caused "institutions for the care and cure of the mentally sick to degenerate into little more than concentration camps" [45:102]. Patterns of systemic abuse, neglect, and suffering were uncovered, including "scores of deaths of patients following beatings by attendants", "a starvation diet, often dragged further below the low-budget standard", and overcrowding of men, women, and children into "hundred-year-old firetraps" [45:102]. This was also true for the Missouri state hospitals, the focus of the present research. Historical research on these institutions using state documents and newspapers was undertaken to contextualize and better understand the fracture results.

Many of the hospitals in which the individuals in this research were confined were established in Missouri between the mid-19th to mid-20th century. One such institution was the St. Louis State Hospital (also known as the City Sanitarium), which opened in 1869 and still operates as the St. Louis Psychiatric Rehabilitation Center. The City Infirmary, or St. Louis Chronic Hospital, was "supported by city taxation for the benefit of the indigent" and was "an example of the type of care available to a significant number of chronic sick" when they could "no long be cared for by other public or private facilities" [46:4]. State Hospital No. 1, which operates today as Fulton State Hospital for the Criminally Insane, was Missouri's first mental asylum and the first to open west of the Mississippi River in December of 1851 [47]. State Hospital No. 2 in St. Joseph followed in 1874 [47]. Nevada State Hospital (State Hospital No. 3) opened in 1889 [47]. The last facility, State Hospital No. 4 (Farmington State Hospital) in Farmington, began accepting patients in 1903 [47].

Morgue records and death certificates of the individuals examined in this study revealed that nine of the perimortem hip fractures examined were preventable (as they were recorded as accidents) and afflicted mostly females. An 82-year-old female, referred to here by both her initials and her Terry Collection (TC) number, AH (TC 639), suffered a fracture of the right femoral neck when she was "Struck down to [the] floor by another patient" at the St. Louis State Hospital September 1930. Her death certificate describes her death as "accidental" and indicates that she died two weeks later from post-fracture complications [48]. AL (TC 1496R) broke her left hip when she "fell from [her] wheel chair at [the] City Infirmary" in February of 1938; her death certificate identifies the principal cause of death to be "fracture of neck of left femur" and notes that she died from complications 12 days later [49]. EW (TC 1326R) fell on the floor of the St. Louis City Infirmary in August of 1943 "while attempting to remove a chair from in front of her door" and died three months later due to a combination of post-fracture complications and chronic myocarditis [50]. One wonders why there was a chair in front of her door. SB (TC 349RR) broke her left femur in April of 1944 "when she fell to [the] floor at City Infirmary while going to the bathroom" [51]. She died three weeks later, with a pulmonary embolism associated with the femoral fracture listed as the principal cause of death. LF,

who was institutionalized for over 15 years in Missouri State Hospital No. #1, fell and fractured her hip in August of 1948 [52]. Her death was also ruled an accident. RP "slipped and fell to [the] floor" in "Ward C-2" at the St. Louis State Hospital in January of 1949. She died 12 days later [53]. In September of 1956, OC (TC 14RR) fell on the floor in Missouri State Hospital No. 3 and died from fracture-related complications [54]. It should be noted that none of these institutionalized individuals, or the others with perimortem fractures, had any evidence of surgical intervention associated with their trauma. This contrasts with another female individual from this study, who was not institutionalized, and had surgical intervention to repair her hip fracture.

Each of these fracture-inducing incidents described above clearly illustrates that individuals were not experiencing trauma only due to their old age. On the contrary, their fractures, which could have been prevented, as they were in a hospitalized and institutionalized setting, resulted from broader, overlapping structural issues associated with underfunding, understaffing, overcrowding, and poorly trained attendants, issues which continue to be commonplace in modern nursing home and assisted living facility settings.

The St. Louis State Hospital had the second highest rates of hip fractures in this study (Fig 4). In 1949 it had a "normal capacity of 2,370" but housed 3,475 patients, resulting in a physician/patient ratio of 1:435 [55]. This violated Missouri state laws and American Psychological Association (APA) regulations of "one physician for every 150 patients" [55]. Fifteen nurses served the entire hospital, or one for every 240 patients [55]. The attendant/patient ratio was 1:15, almost twice the mandated 1:8 [55]. By 1951 the St. Louis State Hospital had 3450 patients, but only 12 doctors, 13 trained nurses to work 47 wards, and 500 attendants, "whose only requirement for employment is an eighth grade education" [56], meaning they may only have been educated until the age of 13 or 14. Many patients were "neglected" and "sat on benches or rocking chairs without even proper exercise. Two and three beds were crowded into tiny rooms hardly large enough for one" [56]. The floor boarding was also precarious, defective, and induced falls. According to the St. Louis Post-Dispatch [57]: "The narrow wooden strips, separated by decay, bend downwards or bulge. Elderly patients have suffered severe falls."

The City Infirmary, which had the highest rates of fractures in this study, struggled with similar issues. Originally established for aged paupers with only chronic conditions, by the 1940s it was an overflow institution for patients not accepted by the St. Louis State Hospital [57]. In 1959, the City Infirmary, which could only support 1000, housed 1424 chronically ill patients. Of these, 278 were mentally ill, with 194 "in two wards where only three attendants are on duty daily from 2:30p.m. until 7:30 o'clock the next morning" [58:1]. After 1955, most of these patients were sent to the Infirmary after being "refused admittance to the St. Louis State Hospital because of overcrowding" [58:4A]. Furthermore, prior to 1946 many of the St. Louis State Hospital's patients were transferred to the Infirmary when they became aged [58]. This influx resulted in 121 "mental cases" considered "by hospital authorities to be dangerous and capable of violence" [58:1]. There were "many incidents in which younger patients have attacked older ones, who are unable to defend themselves" [58:1]. In a situation like this, three attendants would be incapable of quelling major disturbances or preventing violent interactions that resulted in fractures.

The City Infirmary also dealt with infrastructure issues caused by poor state funding. In 1942, the institution's metal fire escapes had dangerously corroded and loose steps, making them a safety hazard for the aged inmates [57]. Patients spending time in the yard below were often "struck by pieces of metal which have fallen from the escapes" [57:5A]. Overcrowding was so severe that there was only "one wash basin for every 32 persons, and one bathtub for every 41 persons" [57:5A]. The infirmary also had improperly insulated wiring, hanging light

sockets in areas where "the floor is likely to be wet", poor roofing with flooding in the laundry room, obsolete equipment that was hazardous to patients, and crumbling concrete steps [57:5A].

Although most of the institutionalized individuals in this study came from the City Infirmary and St. Louis State Hospital, structurally all of Missouri's public mental institutions were plagued with the same issues. Fulton State Hospital was, and continues to be, overcrowded and understaffed [47]. These conditions existed due to "lack of adequate [state] appropriations, which. . .stems from the public attitude. Only $1.926 per day" was spent per patient; the then APA required minimum was $5.00 [56]. Furthermore, accounts from the early 1940s illustrate the abuse patients experienced in Missouri state institutions whilst under the care of poorly trained, and uncaring attendants [45, 59]. For example, in 1945 Missouri's Nevada State Hospital No. 3 was investigated for the death of 67-year-old male patient Cordell Humphries, who had been so severely beaten that he suffered brain trauma [45, 60]. Night attendant Massey E. Cloninger was charged, found guilty of manslaughter, and sentenced to "five years in the state penitentiary" after two patients "testified that they saw Cloninger knock Humphries down, kick him and beat him with a strap" [60:6].

While this analysis focuses upon institutionalized individuals in 1940s Missouri, issues of fracture trauma, structural violence, and institutionalization are neither exclusively historical nor exclusively American issues. A series of critical investigative reports undertaken by *Miami Herald* journalists in 2011 on the state of Florida's assisted living facilities revealed the most vulnerable housed in these facilities are "routinely abused and neglected to death" [61]. For example, over the span of a month in 2008, employees at a Deerfield Beach, Florida, assisted living facility discovered a resident "sprawled on the floor, her body covered in cuts and bruises" [61]. Warnings from the staff nurse about the nonagenarian's constant falls were ignored by the facility's staff, until her eleventh fall when a Broward County's sheriff's deputy found her "lying in a puddle of blood in a locked room, screaming for help" [61]. Her hospital admission revealed a lacerated nose, black eyes, and "a fractured neck" [61]. At a different assisted living facility in Kendall, Florida, the family of a 74-year-old female was notified that she fell in the shower but obtained no injuries. When a relative visited later that evening, she was curled in pain. Upon admission to the hospital, the female was unconscious, "with blackened feet and deep bruises inexplicably circling her legs" [61]. She died two days later; an investigation revealed that her injuries were not associated with a fall in the shower. A caretaker had strapped her in constraints so tightly that it impeded her circulation, resulting in a fatal blood clot.

Other international examples reveal critical issues of neglect associated with hip fractures suffered while institutionalized. Forensic analysis of the skeletal remains of a 76-year-old psychiatric inpatient from Mexico, whose body was accessioned at a medical school in 1996, revealed an intertrochanteric hip fracture with surgical intervention and both healing and healed rib fractures. The broken surgical plate, potentially due to inadequate postoperative care in the institution, resulted in degenerative changes to the acetabulum, which would have severely limited mobility and caused significant pain [62]. A 2014 case report from the UK describes a 79-year-old nursing home resident with dementia who experienced an unwitnessed fall, which resulted in a proximal humeral and bilateral femoral neck fractures. Despite the individual's known medical history, which included dementia and Parkinson's, her facility was "unable to provide any information about her fall, simply saying she had been found on the floor of her bedroom by a nurse" [63:2]. In Oakville, Canada, a nursing home resident died in 2017 with spiral fractures in both femora; the Office of the Chief Coroner deemed her death to be "undetermined" noting that the fractures would have been caused by a "significant rotational force" [64].

## Conclusion

Contextualized skeletal analyses can play a critical social justice role by demonstrating the chronicity of structural issues affecting vulnerable individuals. The discipline seeks to reconstruct how environmental and social stressors impacted biological health by examining multiple data sources including biology, anatomy, history, and modern clinical studies. These skills allow bioarchaeologists and paleopathologists to apply their "knowledge of deep time to contemporary and future challenges that face humankind", including epidemics, inequality, and violence [65:39]. This study demonstrated how structural issues tied to poor patient care in a historic institutionalized setting resulted in physical harm to patients in the form of fatal fractures. Unfortunately, these same factors and patterns persist into the present day amongst many institutionalized groups and continue to result in harm and poor living conditions for vulnerable individuals. We hope that this research is a call to action, impacting future policies entrenching the care of institutionalized individuals as a human right, through dismantling both structural violence and structural apathy.

## Acknowledgments

Thanks to David R. Hunt and Douglas Owsley of the Smithsonian for access to and assistance with the RJTC. Many thanks are also owed to the late Dr. Antonio de la Cova. Thank you to the reviewers for their supportive and helpful comments.

## Author Contributions

**Conceptualization:** Carlina de la Cova, Madeleine Mant, Megan B. Brickley.

**Data curation:** Carlina de la Cova.

**Formal analysis:** Carlina de la Cova, Madeleine Mant, Megan B. Brickley.

**Funding acquisition:** Carlina de la Cova.

**Investigation:** Carlina de la Cova, Madeleine Mant, Megan B. Brickley.

**Methodology:** Carlina de la Cova, Madeleine Mant, Megan B. Brickley.

**Project administration:** Megan B. Brickley.

**Resources:** Carlina de la Cova.

**Supervision:** Megan B. Brickley.

**Visualization:** Carlina de la Cova.

**Writing – original draft:** Carlina de la Cova, Madeleine Mant, Megan B. Brickley.

**Writing – review & editing:** Carlina de la Cova, Madeleine Mant, Megan B. Brickley.

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
