## [Decision Letter · Decision Letter 0]

9 Jan 2023

PONE-D-22-31260Structural violence and institutionalized individuals: a bioarchaeological perspective on a continuing issuePLOS ONE

Dear Dr. Mant,

Thank you for submitting your manuscript to PLOS ONE. After careful consideration, we feel that it has merit but does not fully meet PLOS ONE’s publication criteria as it currently stands. Therefore, we invite you to submit a revised version of the manuscript that addresses the points raised during the review process.

Comments from the Academic Editor:

Dear Dr. Mant,

We appreciate you submitting your manuscript to PLOS ONE and thank you for giving us the opportunity to consider your work.

I have completed my evaluation of your manuscript, which has been reviewed by four highly qualified reviewers all of whom agree it is worth to be published in PLOS ONE. Nevertheless, they have suggested some changes that will help to improve the paper.

Therefore, I invite you to resubmit your manuscript after addressing the reviewers’ comments below. When revising your manuscript, please consider all issues mentioned in the reviewers' comments carefully: please, outline every change made in response to their comments and provide suitable rebuttals for any comments not addressed. Please, note that your revised submission may need to be re-reviewed.

PLOS ONE values your contribution and I look forward to receiving your revised manuscript.

Yours sincerely,

Dr. Olga Spekker

We look forward to receiving your revised manuscript.

Kind regards,

Olga Spekker, Ph.D.

Academic Editor

PLOS ONE

Journal Requirements:

2. In your manuscript, please provide additional information regarding the human remains used in your study. Ensure that you have reported accession numbers and complete repository information.  For more information on PLOS ONE's requirements for paleontology and archaeology research, see https://journals.plos.org/plosone/s/submission-guidelines#loc-paleontology-and-archaeology-research.

Reviewers' comments:

Reviewer's Responses to Questions

**Comments to the Author**

1. Is the manuscript technically sound, and do the data support the conclusions?

Reviewer #1: Partly

Reviewer #2: Partly

Reviewer #3: Partly

Reviewer #4: Yes

2. Has the statistical analysis been performed appropriately and rigorously? 

Reviewer #1: Yes

Reviewer #2: No

Reviewer #3: No

Reviewer #4: Yes

3. Have the authors made all data underlying the findings in their manuscript fully available?

Reviewer #1: Yes

Reviewer #2: No

Reviewer #3: Yes

Reviewer #4: Yes

4. Is the manuscript presented in an intelligible fashion and written in standard English?

Reviewer #1: Yes

Reviewer #2: Yes

Reviewer #3: Yes

Reviewer #4: Yes

5. Review Comments to the Author

Reviewer #1: This is an interesting topic and a valuable contribution to the study of issues surrounding institutional settings. It adds to a growing number of papers which use bioarchaeological methods to shed light on various aspects of life and death at such institutions. The paper aims to investigate hip fractures in the context of structural violence.

It is well-written and the literature is appropriate. The discussion contains enlightening insights into the living conditions and individual fates of inmates at 19/20th century hospitals and institutions which can deepen our understanding of possible events and circumstances leading to hip fractures in such settings, and the ways in which institutionalization can increase the fracture risk.

There is still one main issue that should be addressed to strengthen the paper, apart from that I only have some minor comments.

1) To examine if and how structural violence may be responsible for a higher rate of hip fractures in the institutionalized sample, I feel that it is not sufficient to compare the overall rate of hip fractures in a non-institutionalized vs. an institutionalized sample. Age and sex are known to have a great influence on the incidence and mortality of hip fractures (as is rightly pointed out in the Introduction). However, the paper does not provide information and a discussion about the demographic composition of the two groups (also, it is not indicated what the selection criteria (line 88/89) for the 600 individuals were). The results are presented in a very brief way. More details about the observed fractures (e.g. perimortem/premortem) and the affected individuals would be useful. In the present form of the manuscript, I feel that the conclusion (structural violence as the main cause of the higher fracture rate in the institutionalized group) is not yet fully supported without critically exploring and discussing other possible factors at play.

2) The fracture rates in the abstract (3.7%/11.7%) seem to contradict those provided in the table (page 8, 3.8%/9.8%) and need to be checked.

3) Line 98/99: fracture rate differences between institutions could possibly also imply a different age/sex/ethnic composition of the inhabitants at various institutions, in addition to factors tied to structural violence?

4) Line 135: Is the number of those who had observable femora identical with the total sample size? That does not become entirely clear. If the presence of observable femora was a selection criterion, it would also be useful to state this in the Materials section

Reviewer #2: Dear author

I have genuinely appreciated this manuscript. However, I have recommended major revisions because I have highlighted some issues with the data/results presentation that may require processing time and will impact the final manuscript. But this is a very interesting approach to the discussed topic, and I look forward to reading the final version.

I have linked some comments to specific sections of the manuscript. See below:

_line 78-81: If the focus is the analysis of human remains, this approach has significant limitations. Please consider adding something on the limitations of assessing past behavior and health via the analysis of human remains.

Material and Methods:

_Please provide more details on the sample demographic profile: biological sex, age at death, time /period of death, etc;

_line 90: "and hip fracture prevalence was examined. Institutional REB assessment was not required for this research" - clarify the use of the word REB 1st time used as readers not familiar with it may question what it is;

_Please provide detailed information on how the paleopathological assessment of the remains was undertaken;

_Did you use the biographical data on the records to infer biological sex and age at death; or did you undertake the analysis of the remains, and then compare them to the records? Some collections are known to have incorrect information associated to the remains - please add this information;

_the criteria used to select the individuals was based on whether they were institutionalized or not. (line 95-99). However, when conducting a paleopathological analysis it is also important to exclude cases of individuals with underlying pathologies that may have favor traumatic events - was this taken into consideration? Please clarify;

RESULTS:

_I know it will not change the outcome, but it would be interesting to have additional data on which femora had a fracture (left / right), and were there any bilateral cases? Did individuals exhibit multiple traumas? Why was this not considered?

_please provide information on the healing process of the various fractures. It is essential to have information on the healing stages of the fractures - were these healed entirely at the time of death? In these cases, unless you have undertaken complementary microscopical analysis you cannot state, with certainty, when the fracture occurred - it may have been years before the individual was institutionalized - this need to be considered in your analysis;

_alongside the missing information on the healing stage of the fractures, data on evidence of treatment/reduction is also missing. Was there any?

_there is missing information on fractures data distribution per sample sex and age at death - How do the fractures relate to biological sex and age at death? Both these variables may contribute to the frequency of cases. Imagine that the institutions with more factures are those with elderly individuals, primarily women. We may have a hidden bias related to underlying diseases such as osteoporosis. This needs to be incorporated into the analysis;

_you should complement the table with a more detailed overview of the data comparing institutionalised versus non-institutionalised, per sex, age at death, healing stage of the lesions, and time of death comparing to a traumatic event. This last observation would provide support to the discussion;

DISCUSSION

_line 162: the use of the term bioarchaeology is not adequate. A bioarchaeological analysis relates to the study of remains excavated from an archaeological context: which is not the case here. I would suggest the use of the term paleopathological analysis - of which trauma assessment in past individuals is a major component;

_lines 211/213: as previously commented, some fractures may have underlining causes - older women due to osteoporosis may have more breakable bones: this is illustrative of one of the limitations of the paleopathological analysis, we have access only to a partial story. Please note that my observation DOES not minimize the study, its focus, and its findings; it is just something else to consider when interpreting the results. Some individuals could have been frail due to cumulative years of illness, undernutrition, etc., which would favor the fractures and complications;

_line 214 - "suffered an "accidental" fracture of the" - I understand the use of ""when assessing the events that led to the fractures - but the fact is that we cannot know what happened. Some of the events I have read (below) were accidental. So, rather than using"", please consider stating why you have targeted these specific cases and how they are illustrative of your main argument. Is the word accidentally used in the original description/report? If so, state that and then explain why you question it being described as accidental.

_when describing the individual cases (line 211 onwards), there is an emphasis placed on the time of death in relation to the traumatic event- e.g., line 220 "died three months later [34].", line 222 "She died three weeks later." - I would like to know there anything on the death certificates and morgue records that related the incident/event of trauma as being the cause of death? This information should be included in the case description. When reading the very brief case descriptions, the emphasis you have placed on the event/accident in relation to death is suggestive of a cause /effect. This is something you mention below - line 227 "Each of these fracture-inducing incidents resulted from broader, overlapping structural issues associated with underfunding, understaffing, overcrowding, and poorly trained attendants, issues..." For this reason, adding more information on the cause of death is essential;

_also, I was wondering: in the cases where death occurred shortly after death, why not describe the bone changes observed; the same applies to the cases that had occurred the longer - if the aim of the manuscript not to serve as a reference linking the timing of trauma with the type of lesions observed in the remains is essential;

_line 270 there is a typo "which. . .stems from the"

A proper discussion of your results is missing: age breakdown, sex breakdown, stage of healing of the trauma assessed, and how these relate to the time of death, etc. And how are your results comparable and/or add significantly to the many other situations you mentioned.

CONCLUSION

_line 312, again, the use of the term bioarchaeology is not appropriate. The emphasis should be placed on paleopathology. Whether we are studying the remains of individuals excavated or amassed/ gathered into collections, the emphasis is on the paleopathological analysis when assessing health and behavior, e.g., trauma or others. Please, replace it throughout the manuscript.

Reviewer #3: Review of PONE-D-22-31260

Structural Violence and Institutionalized Individuals

de la Cova, Mant, Brickley

This manuscript explores the concept of structural violence manifesting in the form of hip fractures among institutionalized people in the 19th century and whose remains are today part of the Terry Anatomical Collection. Results demonstrate a significantly higher prevalence of hip fracture among institutionalized individuals. The ms. consideres how structural violence and structural apathy undercut prevention of injury of specific, marginalized categories of people. The work highlights many of these same factors still exist today in institutions of so-called “care” today. The authors demonstrate how a bioarchaeological perspective sheds light on the pervasiveness and time depth of this issue, and seeks to help stimulate public policy changes to “entrench the equitable care of institutionalized people as a human right.”

Overall, I found this ms. to be excellent. It is very well-written, well-conceptualized and designed, and it contains a unique and compelling call by bioarchaeologists in the effort to reduce inequality and suffering in the here-and-now. I believe this ms. is close to being ready for publication, but I would kindly ask the authors to consider making a few changes.

My one principal critique involves the chi-square analysis that determined a significant difference in fracture rates. Especially for traumatic injury, age must be controlled for. If age distributions in the “absent” cohort are even moderately different than the “present” cohort, the chi-square values could reflect a function of age-at-death rather than actual fracture prevalence. This involves a statistical probability of risk related to the passage of time and exposure to risk factors. The more years one has accumulated, the more time has passed to be exposed to traumatic injury, and the greater likelihood of fracture occurring and multiple fractures accumulating. Personally, I think the pattern described by the chi-square results are real and will hold true, but I wish the authors’ work not to receive any undue critiques. Odds ratios are THE purpose-built tool for this situation. Divide each cohort into three or more age categories, calculate the odds ratio for each age group, and from that, calculate the common odds ratio for true prevalence. Then, a chi-square tests for significance. This is very important to do and will make the findings bulletproof against any criticism or doubt.

Minor points:

1. Please consider changing the use of the abbreviation ALF. Every time I read that, I was reminded of a certain late 1980s sitcom, and not an assisted living facility. Simply using the term “assisted living facility” in the text will also help achieve greater clarity for the reader in general.

2. In bioarchaeology, we often say the field is “uniquely positioned” for this and that. This statement is quite factual, but the use of that phrase is becoming somewhat repetitive in many publications and perhaps approaching a cliché. Please consider using different wording to convey the same meaning.

3. Please elaborate on methods – how hip fractures were visually observed and detail on recording protocols, especially.

4. I know de la Cova and Mant to be exceptionally competent bioarchaeologists to the point I am not worried about the following issue – but explain to other readers why interobserver error is not a factor in basic observation and reporting of these frxs.

5. Table 1 heading – for clarity, perhaps consider changing heading “Absent (%)” to “Fracture Absent (%) and changing “Yes (%)” to “Fracture Present (%).” I might also think rewriting the data as a percentage out of a whole to be easier format for most readers, such as “302/314 (96.2%).”

6. Page 9, ln. 199: simplify wording and length of that sentence

7. Page 14, ln. 302- limited mobility and quite likely, chronic, unattended/neglected pain.

Overall, this is an excellent and important ms. I wish the authors success in their revisions and look forward to seeing this work in PLoS ONE.

Reviewer #4: This is an excellent and very powerful paper. The authors' have managed a difficult and emotive topic with empathy and skill; the emotional labour involved must have been very great.

Their drawing together of the bioarchaeological data and the primary source information is impressive, with the extracts from reports and newspapers used to shine a light on these marginalised histories. The paper also serves to remind readers that the people whose curated remains form part of known collections are not just 'medical specimens' - they suffered and endured. The exemplary work fulfils their aim to show that these structural issues have time-depth, and it is certain that the paper will be taken-up by colleagues in the fields of medical history and social justice.

As the manuscript is excellent, there are only a few minor comments.

Data availability - describe where the data may be found: this is likely to be an over-sight. This section includes information about the Hamann-Todd collection, which were not included in this paper (as per lines 88-89).

77: citations 17-22 - please see comment for line 108.

88: please insert 'adult' before 'individuals', as many readers may not know that the RJTC only includes adult individuals. For clarity, please also give the age-range for 'adult', such as >18 years old.

89-90: just for clarity, and to create a better link between this section (also lines 128-9) and the results (lines 135-6), please state that you were only able to include individuals who had observable femora.

100-105: given that the journal is not specific to bioarchaeology, it is suggested that a table is created to summarise the cited methods. This would help to further support the paper's aims and objectives, demonstrate the robustness of the methods used, and underscore the value of including such data alongside post-mortem information. It is also suggested that it would be worth (briefly) highlighting that there are often discrepancies (for a variety of reasons) between autopsy reports and osteological analyses, and why it is necessary to draw on both datasets (e.g. Cappella et al. 2014a, doi: 10.1016/j.forsciint.2014.09.003; 2014b doi:10.1111/1556-4029.12539).

106-107: the P value needs to be given.

108: at lines 211-30, you provide incredibly moving osteobiographical information about three women and raise the important finding that the majority of observed hip fractures occurred in females. This raises the question as to why the fracture data are not given by sex and age cohorts. Given the important work by Prof. de la Cova about how the trauma observed in known collections intersects with other inequalities (e.g. racism), would it be possible to cite these here or in the section about the collection (lines 110-131), so readers (not familiar with the works) can see that these factors have been considered and addressed elsewhere. It is understood that these are cited on line 77, but it is felt that they could contribute more clearly/powerfully if they were not 'tucked-away' in the citations.

Line 151 and Fig 1: caption for Fig 1. It may very well be my (post-covid) reading of the caption and linking the information from lines 119-122 and 144-150, but it would be very helpful if in Fig 1, the establishments were people were institutionalized could be identified - by a * or similar. Otherwise, to understand the Figure and how it relates to Table 1, the reader has to do a certain amount of cross-checking with lines 119-22 and 144-150 to see which institutions would have enabled people's bodies to be anatomized and curated without their consent. If this has not been picked-up by other reviewers, please ignore, as just could be my post-covid brain fog!

238: would it be possible to add that 8th graders are between 13-14 years old, as many readers will not be familiar with the North American school system.

249-50: just a suggestion! To move 'before 1955' to after [42:1], so the sentence reads 'After 1955, most of ... Infirmary'

260-1: 'were "struck by pieces of metal" ' - insert 'often' after 'were', otherwise it can read as if all patients, whenever they went to the yard, were always struck by debris.

6. PLOS authors have the option to publish the peer review history of their article (what does this mean?). If published, this will include your full peer review and any attached files.

Reviewer #1: No

Reviewer #2: No

Reviewer #3: No

Reviewer #4: No

---

## [Author Response · Author response to Decision Letter 0]

18 Jul 2023

Thank you to all four reviewers and the Associate Editor for their feedback. All three authors have approved the following changes to the manuscript. 

Reviewer #1

This is an interesting topic and a valuable contribution to the study of issues surrounding institutional settings. It adds to a growing number of papers which use bioarchaeological methods to shed light on various aspects of life and death at such institutions. The paper aims to investigate hip fractures in the context of structural violence. It is well-written and the literature is appropriate. The discussion contains enlightening insights into the living conditions and individual fates of inmates at 19/20th century hospitals and institutions which can deepen our understanding of possible events and circumstances leading to hip fractures in such settings, and the ways in which institutionalization can increase the fracture risk.

Response: Thank you to the reviewer for the positive feedback on our research.

There is still one main issue that should be addressed to strengthen the paper, apart from that I only have some minor comments.

1) To examine if and how structural violence may be responsible for a higher rate of hip fractures in the institutionalized sample, I feel that it is not sufficient to compare the overall rate of hip fractures in a non-institutionalized vs. an institutionalized sample. Age and sex are known to have a great influence on the incidence and mortality of hip fractures (as is rightly pointed out in the Introduction). However, the paper does not provide information and a discussion about the demographic composition of the two groups (also, it is not indicated what the selection criteria (line 88/89) for the 600 individuals were). The results are presented in a very brief way. More details about the observed fractures (e.g. perimortem/premortem) and the affected individuals would be useful. In the present form of the manuscript, I feel that the conclusion (structural violence as the main cause of the higher fracture rate in the institutionalized group) is not yet fully supported without critically exploring and discussing other possible factors at play.

Response: We have undertaken a more detailed description of the results (starting at line 350) to provide the requested information, including new tables that display the demographic composition (social race, estimated sex, age) of the whole sample and the institutionalized vs. non-institutionalized groups. The selection criteria for the 600 individuals is outlined in greater detail in the Methods (starting at line 163). We have provided further details regarding the perimortem fractures (Table 9, line 468).

2) The fracture rates in the abstract (3.7%/11.7%) seem to contradict those provided in the table (page 8, 3.8%/9.8%) and need to be checked.

Response: Thank you for noticing this! The tables have been updated and the abstract adjusted accordingly (lines 22, 24, 25).

3) Line 98/99: fracture rate differences between institutions could possibly also imply a different age/sex/ethnic composition of the inhabitants at various institutions, in addition to factors tied to structural violence?

Response: This is a great point and one that could be investigated in further research if intake records were made available for the various institutions noted in this research to help untangle the effects of age/sex/ethnic group. The more detailed results section provides a preliminary look at these effects. The historical research included in this current manuscript emphasizes how conditions in the various Missouri institutions were universally poor for those incarcerated.

4) Line 135: Is the number of those who had observable femora identical with the total sample size? That does not become entirely clear. If the presence of observable femora was a selection criterion, it would also be useful to state this in the Materials section

Response: Every individual assessed had both femora. We have noted this more clearly in the Methods section (line 206).

Reviewer #2: 

Dear author, I have genuinely appreciated this manuscript. However, I have recommended major revisions because I have highlighted some issues with the data/results presentation that may require processing time and will impact the final manuscript. But this is a very interesting approach to the discussed topic, and I look forward to reading the final version.

Response: Thank you for taking the time to review the manuscript. We appreciate the points you have raised and have responded to them below.

I have linked some comments to specific sections of the manuscript. See below:

_line 78-81: If the focus is the analysis of human remains, this approach has significant limitations. Please consider adding something on the limitations of assessing past behavior and health via the analysis of human remains.

Response: This is an important point given the broad readership of PLOS One. To maintain the balance of the introduction, we have not gone into details (where would you stop!) but provided a note that care is required and that there are limitations (lines 78-79). The three recent references highlight the value of information obtainable from human remains alongside limitations such as decreasing accuracy in estimating fracture timing with distance from the event. The Plomp et al. chapter [22], which introduces a book on interpreting behaviour from the skeleton, contains a short history and excellent additional references.

Material and Methods:

_Please provide more details on the sample demographic profile: biological sex, age at death, time /period of death, etc;

Response: We have added detail to this section, noting the male/female makeup of the sample (Table 1, line 169), the age-at-death of the sample broken into ten-year age cohorts (Table 2, line 356; Figures 1-3, lines 356, 377, 378), and added details about the amassment of the RJTC (lines 120-144) to contextualize the time of death and collection. 

_line 90: "and hip fracture prevalence was examined. Institutional REB assessment was not required for this research" - clarify the use of the word REB 1st time used as readers not familiar with it may question what it is;

Response: We have adjusted the text to read research ethics board (line 294).

_Please provide detailed information on how the paleopathological assessment of the remains was undertaken;

Response: We have noted the standard osteological methods used to assess age and sex (references 35, 36). We have added a description of how perimortem trauma was assessed in the remains (lines 208-281).

_Did you use the biographical data on the records to infer biological sex and age at death; or did you undertake the analysis of the remains, and then compare them to the records? Some collections are known to have incorrect information associated to the remains - please add this information;

Response: Osteological analyses were conducted to assess sex and age at death. These results were then compared to the documentary evidence available. We have added further details in the methods to describe this process (lines 283-290).

_the criteria used to select the individuals was based on whether they were institutionalized or not. (line 95-99). However, when conducting a paleopathological analysis it is also important to exclude cases of individuals with underlying pathologies that may have favor traumatic events - was this taken into consideration? Please clarify;

Response: In the present study we did not exclude individuals with underlying pathological conditions because the focus of this work was on the difference in the institutionalized vs. non-institutionalized cohorts. It is possible that institutionalization itself had an effect on the overall bone health/quality of these individuals (see citation 27 regarding individuals with osteomalacia in an asylum setting). 

RESULTS:

_I know it will not change the outcome, but it would be interesting to have additional data on which femora had a fracture (left / right), and were there any bilateral cases? Did individuals exhibit multiple traumas? Why was this not considered?

Response: The focus of the present manuscript is on perimortem hip fractures that have associated medical and/or death documentation as a means of examining risk of institutionalization in the past, as perimortem hip fractures are today a risk for institutionalized individuals. Further research, looking at institutionalized individuals with other types of trauma (i.e., multiple, mixture of antemortem and perimortem) would be of interest, but is not our present focus. There were no bilateral cases (this is now noted in the text, line 401). 

_please provide information on the healing process of the various fractures. It is essential to have information on the healing stages of the fractures - were these healed entirely at the time of death? In these cases, unless you have undertaken complementary microscopical analysis you cannot state, with certainty, when the fracture occurred - it may have been years before the individual was institutionalized - this need to be considered in your analysis;

Response: This is an excellent point. The fractures were all examined macroscopically before looking at the documentary evidence. The associated documentary evidence accompanying the RJTC means that we have dates for all of the hip fractures that were observed as well as dates of death. We have added text into the methods to clarify the process of analysis and further details regarding the perimortem (a.k.a. around the time of death) fractures were cross checked with associated medical records and/or death certificates (lines 205-290).

_alongside the missing information on the healing stage of the fractures, data on evidence of treatment/reduction is also missing. Was there any?

Response: In one case (of a non-institutionalized individual) there was evidence of treatment in the form of perimortem screw holes in the femoral fragments. We have included this information in the discussion and noted that there was no evidence of surgical intervention in any of the institutionalized individuals (lines 677-690).

_there is missing information on fractures data distribution per sample sex and age at death - How do the fractures relate to biological sex and age at death? Both these variables may contribute to the frequency of cases. Imagine that the institutions with more factures are those with elderly individuals, primarily women. We may have a hidden bias related to underlying diseases such as osteoporosis. This needs to be incorporated into the analysis;

Response: We have undertaken a more detailed description of the results (starting at line 349) to provide the requested information, including new tables that display the demographic composition (social race, estimated sex, age) of the whole sample and the institutionalized vs. non-institutionalized groups (Table 1, line 169; Table 2, line 356; Table 3, line 3780; Table 4, line 398).

_you should complement the table with a more detailed overview of the data comparing institutionalised versus non-institutionalised, per sex, age at death, healing stage of the lesions, and time of death comparing to a traumatic event. This last observation would provide support to the discussion;

Response: We have added a more detailed description of the demographic composition (social race, estimated sex, age) of the whole sample and the institutionalized vs. non-institutionalized groups as noted in the comment above. We have added text indicating that all hip fractures were cross checked with the associated medical/death record information to confirm the osteological analysis (lines 209-290). Further details from the death records have been added to the discussion to more clearly connect the associated documentary evidence with the individuals with perimortem hip fractures.

DISCUSSION

_line 162: the use of the term bioarchaeology is not adequate. A bioarchaeological analysis relates to the study of remains excavated from an archaeological context: which is not the case here. I would suggest the use of the term paleopathological analysis - of which trauma assessment in past individuals is a major component;

Response: Thank you to the reviewer for these comments. We identify that this paper is undertaking a paleopathological analysis (abstract line 39, line 98). We have removed the reference to bioarchaeology in line 74, 276, and 454 (line numbers from original manuscript) for clarity. We have also adjusted the title (lines 1-3).

_lines 211/213: as previously commented, some fractures may have underlining causes - older women due to osteoporosis may have more breakable bones: this is illustrative of one of the limitations of the paleopathological analysis, we have access only to a partial story. Please note that my observation DOES not minimize the study, its focus, and its findings; it is just something else to consider when interpreting the results. Some individuals could have been frail due to cumulative years of illness, undernutrition, etc., which would favor the fractures and complications;

Response: We agree, thank you for this point. We have added in references [43-45] regarding loss of bone density over time and further analysis regarding the interaction of age cohorts and institutionalization (Table 8, line 455; discussion lines 586-596).

_line 214 - "suffered an "accidental" fracture of the" - I understand the use of ""when assessing the events that led to the fractures - but the fact is that we cannot know what happened. Some of the events I have read (below) were accidental. So, rather than using"", please consider stating why you have targeted these specific cases and how they are illustrative of your main argument. Is the word accidentally used in the original description/report? If so, state that and then explain why you question it being described as accidental.

Response: The word accidental appears in the associated documentary evidence for individual TH639, thus our use of the word accidental is a direct quotation rather than a word we are emphasizing or questioning using scare quotes. For clarity, we have adjusted the wording to state that “Her death certificate describes her death as “accidental” and…” (lines 660-661).

_when describing the individual cases (line 211 onwards), there is an emphasis placed on the time of death in relation to the traumatic event- e.g., line 220 "died three months later [34].", line 222 "She died three weeks later." - I would like to know there anything on the death certificates and morgue records that related the incident/event of trauma as being the cause of death? This information should be included in the case description. When reading the very brief case descriptions, the emphasis you have placed on the event/accident in relation to death is suggestive of a cause /effect. This is something you mention below - line 227 "Each of these fracture-inducing incidents resulted from broader, overlapping structural issues associated with underfunding, understaffing, overcrowding, and poorly trained attendants, issues..." For this reason, adding more information on the cause of death is essential;

Response: Thank you to the reviewer for this point. The death certificates from the Missouri State Board of Health do list causes of death and, in the cases highlighted, identify the hip fractures as the principal causes of death or related to the principal cause of death. For instance, for individual TC 1496, the cause of death reads: “Fracture of neck of left femur, suffered in fall from wheelchair at City Infirmary on Feb 19th.” We have added details regarding the records in reference to the individuals described in the discussion (lines 660-661, 663-665, 667-668, 670-672, 675-676).

_also, I was wondering: in the cases where death occurred shortly after death, why not describe the bone changes observed; the same applies to the cases that had occurred the longer - if the aim of the manuscript not to serve as a reference linking the timing of trauma with the type of lesions observed in the remains is essential;

Response: We are not entirely sure we understand this comment. In all cases the skeletal remains were examined macroscopically for evidence of fracture – in many cases this presented as comminuted fractures, many of which have been described and presented in:

Morgan, B., Mant, M., de la Cova, C., & Brickley, M. Osteoporosis, osteomalacia, and hip fracture: a case study from the Terry Collection. International Journal of Paleopathology. 30: 17-21. https://doi.org/10.1016/j.ijpp.2020.03.004

Mant, M., de la Cova, C., Ives, R., & Brickley, M.B. Perimortem fracture manifestations and mortality after hip fracture in a documented skeletal series. International Journal of Paleopathology. 27: 56-65. https://doi.org/10.1016/j.ijpp.2019.09.002

As noted in comments above, we have added further details on how we determined the presence of perimortem fractures (all of which had associated documentary evidence).

_line 270 there is a typo "which. . .stems from the"

Response: The ellipsis has been put in place to shorten a much longer quotation. This is not a typo (line 765).

A proper discussion of your results is missing: age breakdown, sex breakdown, stage of healing of the trauma assessed, and how these relate to the time of death, etc. And how are your results comparable and/or add significantly to the many other situations you mentioned.

Response: The age and sex breakdown details have been added to the results section, as noted in previous comments. We have added further details regarding perimortem status (Table 9, line 468).

CONCLUSION

_line 312, again, the use of the term bioarchaeology is not appropriate. The emphasis should be placed on paleopathology. Whether we are studying the remains of individuals excavated or amassed/ gathered into collections, the emphasis is on the paleopathological analysis when assessing health and behavior, e.g., trauma or others. Please, replace it throughout the manuscript.

Response: We have removed the term bioarchaeology and/or added paleopathology as a qualifier throughout the manuscript (lines 39, 80, 98, 499, 833).

Reviewer #3: Review of PONE-D-22-31260

Structural Violence and Institutionalized Individuals

de la Cova, Mant, Brickley

This manuscript explores the concept of structural violence manifesting in the form of hip fractures among institutionalized people in the 19th century and whose remains are today part of the Terry Anatomical Collection. Results demonstrate a significantly higher prevalence of hip fracture among institutionalized individuals. The ms. consideres how structural violence and structural apathy undercut prevention of injury of specific, marginalized categories of people. The work highlights many of these same factors still exist today in institutions of so-called “care” today. The authors demonstrate how a bioarchaeological perspective sheds light on the pervasiveness and time depth of this issue, and seeks to help stimulate public policy changes to “entrench the equitable care of institutionalized people as a human right.”

Overall, I found this ms. to be excellent. It is very well-written, well-conceptualized and designed, and it contains a unique and compelling call by bioarchaeologists in the effort to reduce inequality and suffering in the here-and-now. I believe this ms. is close to being ready for publication, but I would kindly ask the authors to consider making a few changes.

Response: Thank you for this positive assessment of our work. We appreciate the points raised and have responded to each below.

My one principal critique involves the chi-square analysis that determined a significant difference in fracture rates. Especially for traumatic injury, age must be controlled for. If age distributions in the “absent” cohort are even moderately different than the “present” cohort, the chi-square values could reflect a function of age-at-death rather than actual fracture prevalence. This involves a statistical probability of risk related to the passage of time and exposure to risk factors. The more years one has accumulated, the more time has passed to be exposed to traumatic injury, and the greater likelihood of fracture occurring and multiple fractures accumulating. Personally, I think the pattern described by the chi-square results are real and will hold true, but I wish the authors’ work not to receive any undue critiques. Odds ratios are THE purpose-built tool for this situation. Divide each cohort into three or more age categories, calculate the odds ratio for each age group, and from that, calculate the common odds ratio for true prevalence. Then, a chi-square tests for significance. This is very important to do and will make the findings bulletproof against any criticism or doubt.

Response: Thank you for this point! We have added odds ratio analyses (Table 7, line 454).

Minor points:

1. Please consider changing the use of the abbreviation ALF. Every time I read that, I was reminded of a certain late 1980s sitcom, and not an assisted living facility. Simply using the term “assisted living facility” in the text will also help achieve greater clarity for the reader in general.

Response: We have adjusted this abbreviation throughout the manuscript (and thank you to the reviewer for the smile!).

2. In bioarchaeology, we often say the field is “uniquely positioned” for this and that. This statement is quite factual, but the use of that phrase is becoming somewhat repetitive in many publications and perhaps approaching a cliché. Please consider using different wording to convey the same meaning.

Response: We have changed this phrasing in the first case (line 76) to say “ideally positioned” and removed it from the opening sentence of the conclusion (line 829).

3. Please elaborate on methods – how hip fractures were visually observed and detail on recording protocols, especially.

Response: We have added details to the methods section regarding how hip fractures were identified, observed, and recorded (lines 187-290).

4. I know de la Cova and Mant to be exceptionally competent bioarchaeologists to the point I am not worried about the following issue – but explain to other readers why interobserver error is not a factor in basic observation and reporting of these frxs.

Response: de la Cova undertook the initial osteological analyses with Mant and Brickley undertaking secondary crosschecks of the death certificates to check. We have added details of these roles (lines 291-294).

5. Table 1 heading – for clarity, perhaps consider changing heading “Absent (%)” to “Fracture Absent (%) and changing “Yes (%)” to “Fracture Present (%).” I might also think rewriting the data as a percentage out of a whole to be easier format for most readers, such as “302/314 (96.2%).”

Response: We have adjusted the tables to use absent/present when referring to the hip fractures. Percentages have been added.

6. Page 9, ln. 199: simplify wording and length of that sentence

Response: We have split the sentence into two and revised the first sentence (now lines 631-635).

7. Page 14, ln. 302- limited mobility and quite likely, chronic, unattended/neglected pain.

Response: Thank you to the reviewer for this observation. We have added a reference to the likely pain (line 810).

Overall, this is an excellent and important ms. I wish the authors success in their revisions and look forward to seeing this work in PLoS ONE.

Reviewer #4: This is an excellent and very powerful paper. The authors' have managed a difficult and emotive topic with empathy and skill; the emotional labour involved must have been very great.

Their drawing together of the bioarchaeological data and the primary source information is impressive, with the extracts from reports and newspapers used to shine a light on these marginalised histories. The paper also serves to remind readers that the people whose curated remains form part of known collections are not just 'medical specimens' - they suffered and endured. The exemplary work fulfils their aim to show that these structural issues have time-depth, and it is certain that the paper will be taken-up by colleagues in the fields of medical history and social justice.

Response: Thank you to the reviewer for this generous assessment of our manuscript. We appreciate the efforts taken in reviewing it and have outlined our responses below.

As the manuscript is excellent, there are only a few minor comments.

Data availability - describe where the data may be found: this is likely to be an over-sight. This section includes information about the Hamann-Todd collection, which were not included in this paper (as per lines 88-89).

Response: Thank you for noting this! This was an oversight and has been revised to focus on the Robert J. Terry Collection.

77: citations 17-22 - please see comment for line 108.

Response: Thank you for this comment. The requested information (as noted above for previous reviewers’ comments has been added.

88: please insert 'adult' before 'individuals', as many readers may not know that the RJTC only includes adult individuals. For clarity, please also give the age-range for 'adult', such as >18 years old.

Response: We have inserted the word adult and the definition (>18 years) on line 108 to make this clear.

89-90: just for clarity, and to create a better link between this section (also lines 128-9) and the results (lines 135-6), please state that you were only able to include individuals who had observable femora.

Response: We have made it clearer that all individuals had both femora and that we examined all observable femora (lines 205-206).

100-105: given that the journal is not specific to bioarchaeology, it is suggested that a table is created to summarise the cited methods. This would help to further support the paper's aims and objectives, demonstrate the robustness of the methods used, and underscore the value of including such data alongside post-mortem information. IIt is also suggested that it would be worth (briefly) highlighting that there are often discrepancies (for a variety of reasons) between autopsy reports and osteological analyses, and why it is necessary to draw on both datasets (e.g. Cappella et al. 2014a, doi: 10.1016/j.forsciint.2014.09.003; 2014b doi:10.1111/1556-4029.12539).

Response: The reference to Capella et al. 2014 has been added [21] with a note that multiple lines of evidence are useful. In response to another reviewer’s comment regarding the limitations of bioarchaeology we have added further literature regarding bioarchaeological analysis and specific bioarchaeological methods [20-22, 35, 36].

106-107: the P value needs to be given.

Response: The p-value of <0.05 has been added (line 346).

108: at lines 211-30, you provide incredibly moving osteobiographical information about three women and raise the important finding that the majority of observed hip fractures occurred in females. This raises the question as to why the fracture data are not given by sex and age cohorts. Given the important work by Prof. de la Cova about how the trauma observed in known collections intersects with other inequalities (e.g. racism), would it be possible to cite these here or in the section about the collection (lines 110-131), so readers (not familiar with the works) can see that these factors have been considered and addressed elsewhere. It is understood that these are cited on line 77, but it is felt that they could contribute more clearly/powerfully if they were not 'tucked-away' in the citations.

Response: Excellent point. The sex and age cohorts have been added, as detailed above for previous reviewers’ comments.

Line 151 and Fig 1: caption for Fig 1. It may very well be my (post-covid) reading of the caption and linking the information from lines 119-122 and 144-150, but it would be very helpful if in Fig 1, the establishments were people were institutionalized could be identified - by a * or similar. Otherwise, to understand the Figure and how it relates to Table 1, the reader has to do a certain amount of cross-checking with lines 119-22 and 144-150 to see which institutions would have enabled people's bodies to be anatomized and curated without their consent. If this has not been picked-up by other reviewers, please ignore, as just could be my post-covid brain fog!

Response: All the institutions listed in the original Figure 1 were places where individuals in the RJTC were institutionalized. We have revised this figure (new Figure 4, line 494) and added some further discussion of the number of individuals with hip fractures institutionalized in each.

238: would it be possible to add that 8th graders are between 13-14 years old, as many readers will not be familiar with the North American school system.

Response: We have added this detail (lines 706-707).

249-50: just a suggestion! To move 'before 1955' to after [42:1], so the sentence reads 'After 1955, most of ... Infirmary'

Response: We have adjusted the wording to that suggested by the reviewer (line 732).

260-1: 'were "struck by pieces of metal" ' - insert 'often' after 'were', otherwise it can read as if all patients, whenever they went to the yard, were always struck by debris.

Response: We have added the word often (line 744).

---

## [Decision Letter · Decision Letter 1]

1 Aug 2023

Structural violence and institutionalized individuals: a paleopathological perspective on a continuing issue

PONE-D-22-31260R1

Dear Dr. Mant,

We’re pleased to inform you that your manuscript has been judged scientifically suitable for publication and will be formally accepted for publication once it meets all outstanding technical requirements.

Kind regards,

Olga Spekker, Ph.D.

Academic Editor

PLOS ONE

Additional Editor Comments (optional):

Reviewers' comments:

Reviewer's Responses to Questions

**Comments to the Author**

1. If the authors have adequately addressed your comments raised in a previous round of review and you feel that this manuscript is now acceptable for publication, you may indicate that here to bypass the “Comments to the Author” section, enter your conflict of interest statement in the “Confidential to Editor” section, and submit your "Accept" recommendation.

Reviewer #1: All comments have been addressed

Reviewer #2: All comments have been addressed

Reviewer #4: All comments have been addressed

2. Is the manuscript technically sound, and do the data support the conclusions?

Reviewer #1: Yes

Reviewer #2: Yes

Reviewer #4: Yes

3. Has the statistical analysis been performed appropriately and rigorously? 

Reviewer #1: Yes

Reviewer #2: Yes

Reviewer #4: Yes

4. Have the authors made all data underlying the findings in their manuscript fully available?

Reviewer #1: Yes

Reviewer #2: No

Reviewer #4: Yes

5. Is the manuscript presented in an intelligible fashion and written in standard English?

Reviewer #1: Yes

Reviewer #2: Yes

Reviewer #4: Yes

6. Review Comments to the Author

Reviewer #1: Thank you to the authors for the outstanding job in addressing the comments. I wholeheartedly recommend "accept" as the conclusions are now fully supported by the data, and can easily be understood after numerous other details have been added for increased clarity. In this form it is an excellent paper!

Reviewer #2: I truly appreciate the time the authors have taken to address / respond to the comments. I have nothing further to add.

Reviewer #4: The revisions to the paper are excellent, and all of the changes suggested have been met or accommodated by the new manuscript. Great work!

7. PLOS authors have the option to publish the peer review history of their article (what does this mean?). If published, this will include your full peer review and any attached files.

Reviewer #1: No

Reviewer #2: No

Reviewer #4: No

---

## [Editor Report · Acceptance letter]

3 Aug 2023

PONE-D-22-31260R1 

Structural violence and institutionalized individuals: a paleopathological perspective on a continuing issue 

Dear Dr. Mant:

I'm pleased to inform you that your manuscript has been deemed suitable for publication in PLOS ONE. Congratulations! Your manuscript is now with our production department. 

Kind regards, 

on behalf of

Dr. Olga Spekker 

Academic Editor

PLOS ONE